

# Rocks of different mineralogy show different temperature characteristics: implications for biodiversity on rocky seashores

Nathan Janetzki[1], Kirsten Benkendorff[2] and Peter G. Fairweather[1,†]

[1] College of Science & Engineering, Flinders University, Adelaide, South Australia, Australia
[2] National Marine Science Centre, Faculty Science and Engineering, Southern Cross University, Lismore, New South Wales, Australia, Lismore, New South Wales, Australia
[†] Deceased.

## ABSTRACT

As some intertidal biota presently live near their upper tolerable thermal limits when emersed, predicted hotter temperatures and an increased frequency of extreme-heat events associated with global climate change may challenge the survival and persistence of such species. To predict the biological ramifications of climate change on rocky seashores, ecologists have collected baseline rock temperature data, which has shown substrate temperature is heterogenous in the rocky intertidal zone. A multitude of factors may affect rock temperature, although the potential roles of boulder surface (upper versus lower), lithology (rock type) and minerology have been largely neglected to date. Consequently, a common-garden experiment using intertidal boulders of six rock types tested whether temperature characteristics differed among rock types, boulder surfaces, and whether temperature characteristics were associated with rock mineralogy. The temperature of the upper and lower surfaces of all six rock types was heterogeneous at the millimetre to centimetre scale. Three qualitative patterns of temperature difference were identified on boulder surfaces: gradients; mosaics; and limited heterogeneity. The frequency of occurrence of these temperature patterns was heavily influenced by cloud cover. Upper surfaces were generally hotter than lower surfaces, plus purple siltstone and grey siltstone consistently had the hottest temperatures and white limestone and quartzite the coolest. Each rock type had unique mineralogy, with maximum temperatures correlated with the highest metallic oxide and trace metal content of rocks. These baseline data show that rock type, boulder surface and mineralogy all contribute to patterns of heterogenous substrate temperature, with the geological history of rocky seashores potentially influencing the future fate of species and populations under various climate change scenarios.

Corresponding author
Nathan Janetzki,
nathan.janetzki@outlook.com

## INTRODUCTION

Rocky seashores are one of the most thermally-variable and stressful habitats on Earth. Rocky seashore substrata exposed to insolation can warm by as much as 10–20 °C while emersed during low tide (*Bertness, 1999*; *Helmuth & Hofmann, 2001*; *Harley, 2008*). This extreme heat and temperature variability presents risks of desiccation and heat stress to intertidal ectotherms that use rocky seashores as habitat (*Connell, 1972*; *Bertness, 1999*), which may challenge the survival and ultimately the persistence of some intertidal species. Examples of the impacts of heat stress on intertidal species include mass-mortality events during heatwaves (*Helmuth et al., 2002*; *Harley, 2008*; *Seuront et al., 2019*), lower survival rates in thermally less-favourable habitats (*Jones & Boulding, 1999*; *Harley, 2008*; *Gedan et al., 2011*; *Lathlean, Ayre & Minchinton, 2013*; *Leal et al., 2020*), and restricted vertical seashore distributions of some species (*Raimondi, 1988*; *Somero, 2002*; *Harley, 2003*).

In response to these deleterious impacts from exposure to extreme heat, mobile intertidal ectotherms can employ behavioural thermoregulation and/or morphological adaptations to minimise the risks posed by desiccation and heat stress (*Pörtner & Farrell, 2008*). One mode of behavioural thermoregulation involves retreating to the underside of boulders at low tide (*Chapman, 2003*; *Chapperon & Seuront, 2011*). Boulder lower surfaces are sheltered from insolation and thus are purported to provide a cooler and more thermally-stable habitat, relative to their thickness (*Huey et al., 1989*), in comparison to sun-exposed boulder upper surfaces (*Chapman, 2003*; *Chapperon & Seuront, 2011*; *Aguilera, Arias & Manzur, 2019*). In the rocky intertidal zone, surprisingly few studies have quantified the temperature characteristics of upper versus lower boulder surfaces, with most studies instead quantifying the temperature characteristics of various sun-exposed versus sun-protected habitats (e.g., *Garrity, 1984*; *Denny et al., 2011*; *Chapperon, Studerus & Clavier, 2017*). It is therefore difficult to ascertain from the published literature the actual magnitude of temperature difference that may exist between the tops and bottoms of boulders.

With predictions of hotter air temperatures and an increased frequency of extreme-heat events associated with global climate change (*IPCC, 2013*), the survival and persistence of some intertidal species is likely to be challenged further. To predict the future fate of species and populations, ecologists have collected baseline rock temperature data (*Helmuth, 1999*; *Denny et al., 2011*; *Judge, Botton & Hamilton, 2011*; *Gunderson et al., 2019*), created heat budget models (*Helmuth, 1999*; *Choi et al., 2019*), investigated how $x$ species is affected by $y$ substrate temperature (*Raimondi, 1988*; *Lathlean, Ayre & Minchinton, 2012*; *Lathlean, Ayre & Minchinton, 2013*; *Lamb, Leslie & Shinen, 2014*), or used biomimetic loggers to investigate how internal body temperatures can be variously affected by environmental temperature (*Helmuth & Hofmann, 2001*; *Seabra et al., 2011*; *Lathlean et al., 2015*; *Seuront et al., 2019*). All of these studies confirm that temperature is an important driving force that can influence the distribution of species on rocky seashores.

Comparing across studies that investigate substratum temperature, a number of rock temperature observations have been made. Rock temperature is affected by its colour (*Raimondi, 1988*; *Judge, Botton & Hamilton, 2011*; *Gunderson et al., 2019*), size and orientation relative to the sun (*Bertness, 1999*; *Chapperon et al., 2016*; *Chapperon,*

*Studerus & Clavier, 2017*). *Marshall, McQuaid & Williams (2010)* reported lighter-coloured sandstone was cooler than darker-coloured ferruginous sandstone. Furthermore, while *Judge, Botton & Hamilton (2011)* failed to identify their rock lithology, they reported that black rock was hotter than white rock. These studies aside, it appears that the role of lithology (rock type) in affecting substrate temperature has been largely neglected to date. It is possible that the specific mineral constituents of different rock types influence their temperature characteristics, although this does not appear to have been investigated for rocky seashore substrata. Investigations of small granitic test cubes have shown that their different mineral constituents can have a temperature range of up to 4 °C (*Gómez-Heras, Smith & Fort, 2006*). Therefore, if ecologists are to accurately predict the future fate of species and populations under various climate change scenarios, how rock types and their specific mineralogies may affect temperature characteristics needs to be better understood.

Studies investigating intertidal rock temperature have also shown that observed substratum temperature is not homogenous (e.g., *Huey et al., 1989*; *Gómez-Heras, Smith & Fort, 2006*; *Helmuth et al., 2006*; *Lathlean, Ayre & Minchinton, 2012*; *Gunderson et al., 2019*; *Leal et al., 2020*). Instead, substratum temperature is heterogeneous, with temperature differences up to 25.5 °C identified between the hottest and coolest locations (e.g., *Huey et al., 1989*; *Denny et al., 2011*; *Lathlean, Ayre & Minchinton, 2012*; *Gunderson et al., 2019*; *Leal et al., 2020*). The scale of these patterns of temperature difference can vary enormously, from many kilometres down to millimetres (*Helmuth et al., 2006*; *Denny et al., 2011*; *Judge, Botton & Hamilton, 2011*; *Lathlean, Ayre & Minchinton, 2012*; *Lathlean, Ayre & Minchinton, 2013*). Identifying the sources of this temperature heterogeneity can be difficult, with rocky seashores not being equal in terms of their physical attributes, with variations in latitude (*Helmuth et al., 2006*; *Lathlean, Ayre & Minchinton, 2014*), shore slope (*Helmuth & Hofmann, 2001*; *Harley, 2008*), azimuth (*Helmuth & Hofmann, 2001*; *Harley, 2008*; *Chapperon, Studerus & Clavier, 2017*), microhabitat features (*Chapperon & Seuront, 2011*; *Judge, Botton & Hamilton, 2011*; *Lathlean et al., 2015*), rock type (*Raimondi, 1988*; *Marshall, McQuaid & Williams, 2010*; *Judge, Botton & Hamilton, 2011*); relative humidity (*Lathlean, Ayre & Minchinton, 2014*), wave splash (*Helmuth et al., 2006*), the timing of low tide (*Helmuth et al., 2002*; *Helmuth et al., 2006*) and microtopography (*Lathlean, Ayre & Minchinton, 2012*; *Choi et al., 2019*) all contributing to temperature heterogeneity.

To investigate how rock temperature characteristics may be affected by rock type, mineralogy, and boulder surface (upper versus lower), a common-garden experiment using boulders of six seashore rock types was established. Three rock temperature characteristics were investigated on the upper and lower surfaces of boulders, which were: the spatial arrangement (i.e., patterns) of temperature on boulder surfaces; the maximum surface temperature; and the surface temperature range (i.e., maxima–minima). We focused on maxima due to extreme temperatures having a greater impact on organism survival and fitness (e.g., *Jones & Boulding, 1999*; *Harley, 2008*; *Gedan et al., 2011*). Sampling was conducted over an 18-month timeframe to investigate whether daily weather conditions and seasonality affected rock temperature. We also quantified the mineralogy of each rock type and investigated which minerals were correlated with rocks showing different

temperature characteristics. Consequently, this experimental approach allowed us to test the following four hypotheses:

(1) boulders of different rock types have different patterns of temperature heterogeneity across their surfaces;

(2) the maximum temperature reflected from the rock surface differs between rock types and between the sun-exposed top and shaded bottom-surface of boulders;

(3) the magnitude of the range in temperature differs between rock types or surfaces; and

(4) rock-related differences in temperature maxima are correlated with their mineral composition.

## MATERIALS & METHODS

### Boulder selection

The geologically-diverse Fleurieu Peninsula of South Australia is comprised of a variety of rock types. Six of these, in the form of small boulders ($n = 6$ boulders per rock type, maximum dimension $\leq 30$ cm), were collected from four seashores (Appendix Fig. A1). From Southport (35°10′S, 138°27′E) boulders of either white fossiliferous limestone or orange fossiliferous limestone were collected, while fossiliferous sandstone that was yellowish brown in colour was collected from Seaford (35°11′S, 138°28′E) (Appendix Figs. A1 & A2). The two limestones and the fossiliferous sandstone had coarse surface textures and complex surfaces that were interspersed by cracks and depressions. From Marino Rocks (35°02′S, 138°30′E) boulders of both purple siltstone and grey siltstone were collected, while quartzite that was greyish to yellow-brown in colour, was collected from O'Sullivan Beach (35°07′S, 138°28′E) (Appendix Figs. A1 & A2). The two siltstones had smooth surface textures and featureless surfaces that generally lacked cracks or depressions, while quartzite had coarse, angular surfaces that also lacked cracks and depressions.

Six boulders of each rock type were collected (total $N = 36$), with boulders specifically selected to span the range of shapes and thicknesses that occurred for each type on each seashore (Appendix Fig. A2). Measurements of thickness were collected for each boulder to be used as a co-variate in statistical analyses. However, as the boulders investigated had only a naturally narrow range of thicknesses (6–14 cm), boulder thickness was never a significant co-variate, and was hence removed from all analyses to improve their statistical power.

### Experimental location

Boulders were transplanted into a paddock on a farm at Kangarilla, which was located approximately 20 km inland from the coast (Appendix Fig. A1). This secure setting was selected to be independent of any confounding variables. For example, had experimentation been conducted at one of the seashores where boulders were sourced, variables such as wave splash, tidal movement, sediment or wrack deposition, shading by cliffs, under-boulder substratum or angle of repose may have affected the temperature of some boulders and not others. Setting boulders into a sandy beach was not attempted due to the boulder losses sustained during a translocation experiment in the same region (*Liversage, Janetzki & Benkendorff, 2014*; *Janetzki, Fairweather & Benkendorff, 2018*). Issues with sand scour or

burial of boulders by sand were also likely on a sandy beach. Moreover, given the large population of the Adelaide region (1.3 million people in 2016), interference with boulders left lying on a beach was considered likely.

## Design of the common-garden experiment

A square plot measuring 3 × 3 metres was excavated to a depth of approximately 10 centimetres (Appendix Fig. A2). The ironstone and soil matrix unearthed was replaced by washed yellow beach sand to simulate substrate matrices where experimental boulders were sourced (Appendix Fig. A2). The location of this plot was selected to ensure it had an east–west orientation (i.e., to follow the movement of the sun) free from any physical obstructions (e.g., buildings, trees) that might shade the plot. Boulders were arranged on the sand matrix in four groups that each contained nine boulders (Appendix Fig. A2). Boulders were arranged such that each group could be sampled without accidently disturbing or shading other groups (Appendix Fig. A2). Boulders were randomly assigned to each group and their location was re-randomised on four occasions. The boulder plot was covered with a tarpaulin to shelter boulders from insolation prior to sampling, although small differences in surface temperature among rock types were identified at time zero on several days sampled (Appendix Figs. A3 & A4).

Rock temperatures were measured on 17 days spread over an 18-month period (Table 1). Each day was targeted for its forecast cloud cover and air temperature to investigate generally how rock temperature may be influenced by seasonality and daily weather conditions. Sampling was completed on days where no rainfall was forecast, as wet surfaces were likely to confound our ability to accurately measure rock temperature (*Lathlean & Seuront, 2014*; *Seuront, Ng & Lathlean, 2018*). Approximately 20 min before the commencement of sampling, the tarpaulin was removed from the boulder plot and each boulder was submersed (<1 min) in a tub filled with seawater. Submersion wetted the boulder surfaces to simulate conditions on the seashore, where boulder surfaces are wet when first emersed by the receding tide. Each boulder was returned to its specific spot in the plot where it was allowed to drain and dry (this took no longer than 5–10 min as water rarely permeated boulder surfaces). There was no evidence of differential patterns of drying over this 5–10 min timeframe among the six rock types investigated.

The surface temperature of each boulder was measured at one-hour intervals, commencing at 0900 h and concluding at 1400 h daily. This five-hour time course was selected to simulate the average length of time boulders on local mid-lower seashores are emersed during a single low tide (NJ unpublished data). The 0900–1400 start and finish times were selected to simulate the timing of summer daytime low tides for seashores on the Fleurieu Peninsula. Sunrise occurred between 0555 h during the height of summer and 0724 h during the depths of winter.

Surface temperature was measured with a Fluke Ti20 thermal imaging camera (Fluke Corporation, Everett). The thermal resolution of this camera was ≤0.2 °C at 30 °C, with accuracy to 2% or 2 °C, whichever was greater. Default camera settings were employed, including emissivity, which was set at 0.95. This default emissivity was applied, even though emissivity may have differed within and among the rock types sampled, as previous

**Table 1 The maximum cloudiness and air temperature recorded during sampling on each day, and the weather condition category that each day was subsequently allocated to based on its maximum cloudiness.** Days are arranged in each weather condition category according to increasing maximum air temperatures.

| Weather condition | Date | Maximum cloudiness (Okta) | Maximum air temperature (°C) |
|---|---|---|---|
| Cloudy | 09/09/2015 | 8 | 15 |
| | 18/09/2015 | 7 | 17 |
| | 06/10/2015 | 7 | 21 |
| | 25/11/2015 | 8 | 30 |
| | 19/12/2015 | 8 | 38 |
| Sunny | 16/07/2016 | 0 | 12 |
| | 10/09/2015 | 0 | 15 |
| | 17/10/2015 | 0 | 22 |
| | 17/04/2017 | 3 | 23 |
| | 02/10/2015 | 0 | 24 |
| | 07/01/2016 | 0 | 26 |
| | 07/02/2016 | 0 | 29 |
| | 06/02/2016 | 0 | 31 |
| | 09/10/2015 | 3 | 33 |
| | 19/11/2015 | 0 | 34 |
| | 08/02/2017 | 0 | 39 |
| | 18/11/2015 | 0 | 40 |

thermography studies in the long-infrared range (9–14 $\mu$m) have shown that the emissivity of dry rock generally ranges between 0.95 and 1 (*Rivard, Thomas & Giroux, 1995*; *Danov, Vitchko & Stoyanov, 2007*; *Cox & Smith, 2011*; *Lathlean, Ayre & Minchinton, 2012*). Surface temperature was captured *in situ*, with archived thermal images processed later. Upper and lower boulder surfaces were imaged separately. In this study, lower surfaces are defined as the underside of the boulder that was in contact with the substratum and thus sheltered from insolation. Upper surfaces are defined as all remaining surfaces that were not in contact with the substratum and were potentially exposed to insolation.

Thermal images were recorded for all upper surfaces first without touching boulders. For lower surfaces, each boulder was briefly flipped upside down, and a thermal image recorded, before the boulder was returned to its original position. Overall, a total 7,344 thermal images were collected. Air temperature and cloud cover were also recorded at one-hour intervals when taking images. Air temperature was measured in the shade to the nearest degree Celsius with a glass thermometer. Cloud cover (i.e., sky condition) was estimated by how many eighths of sky were covered by cloud, which ranged from zero oktas (sunny, no clouds) to eight oktas (sky completely cloudy, no sunshine) (*Li & Lam, 2001*). Each day sampled was assigned to one of two weather condition categories based on their cloud cover. Days where the cloud cover never exceeded 3 oktas were assigned the 'sunny' category, while days where the cloud cover exceeded 3 oktas during sampling were assigned the 'cloudy' category (Table 1).

Archived thermal images were subsequently processed using the InsideIR version 4.0 software (Fluke Corporation, see Appendix for more information). The maximum and minimum temperature of substrata were determined for each replicate surface, and a temperature range (maximum–minimum) for each surface was calculated. The orientation, relative to the sun, for the maximum temperature, was categorised as either occurring on the boulder side nearest the sun, or on any other boulder side. Transects were drawn on images of boulder surfaces from the centre of the boulder side nearest the sun to the centre of the side opposite to quantify millimetre-to-centimetre scale patterns of temperature difference. Analysis of the temperature patterns along transects was undertaken on three sunny and three cloudy days, spanning the range of maximum daily air temperatures sampled. Images for the zero-hour and four-hour exposure times were investigated to look at changes across the day.

## XRF analyses

The mineralogy of each rock type was determined through X-ray *fluorescence* (XRF) dispersion analysis, with separate tests completed for major mineral and trace elemental composition for three samples of each rock type. XRF analysis of each rock sample tested for 11 major minerals and 40 trace elements, with concentrations returned as % and parts per thousand, respectively. For major minerals, approximately one gram of each oven-dried sample (at 105 °C) was accurately weighed with four grams of 12–22 lithium borate flux (*Norrish & Hutton, 1969*). The mixtures were heated to 1,050 °C in a platinum/gold crucible for 20 min to completely dissolve the sample, and then poured into a 32 mm platinum/gold mould heated to a similar temperature (*Norrish & Hutton, 1969*). The melt was cooled rapidly over a compressed air stream and the resulting glass disks were analysed on a PANalyticalAxios Advanced wavelength dispersive XRF system using the CSIRO in-house silicates calibration program. For trace elements, approximately four grams of each oven-dried sample (at 105 °C) was accurately weighed with one gram of Licowax binder and mixed well (*Norrish & Hutton, 1969*). The mixtures were pressed in a 32 mm die at 12 tons pressure and the resulting pellets were analysed on a PANalyticalAxios Advanced wavelength dispersive XRF system using the CSIRO in-house powders program (*Norrish & Hutton, 1969*).

## Statistical analyses and data presentation

Frequencies of occurrence (%) for three patterns of temperature difference were tallied for the upper and lower surfaces of boulders for all six rock types after four hours exposure to insolation. Only data for three cloudy versus sunny days are presented, because each day is not strictly independent of each other.

Line charts plotting temperature dependent variables (maxima, range) versus exposure time were used to rank rock types in descending order from 6 (largest) to 1 (smallest) for their mean temperature range and maxima, for each surface on each day after four hours. The sum of ranks allocated to each rock type was then used to assign an overall rank to each rock type from 6 (largest) to 1 (smallest) for their temperature range and maxima, with the highest ranked rock types having the highest sum of ranks. Upper and lower surfaces

were ranked separately. To quantify changes over time exposed, dependent variables for the upper and lower surface of each replicate boulder after four hours were subtracted from the same dependent variables for the same replicate surface at zero hours. This gave values for the change in maxima and temperature range over four hours, for the upper and lower surfaces of each boulder, on each day sampled. Comparing the magnitude of change over four hours was used in place of formal statistical tests, as measurements were made continuously for the same boulders and thus were not independent. To establish whether dependent variables differed between surfaces, upper-surface dependent variables were subtracted from lower-surface dependent variables, for each replicate boulder, for each exposure time on each day sampled. The resulting difference data were plotted as line charts to visually investigate surface differences over the exposure period.

Analyses were completed using PRIMER v7/PERMANOVA+ (PRIMER-e, Plymouth, UK), with significance set at $\alpha = 0.05$. To test for mineralogical differences between rock types, separate multivariate analyses were completed for major mineral composition and trace element composition, with separate univariate analyses completed for the total content of each major mineral. Untransformed major-mineral data (measured as % composition) were used, while fourth-root-transformed (necessary to down-weight several dominant elements) trace-element data (measured as parts per million) were used. Euclidean-distance resemblance matrices were prepared, and PERMutational Analyses Of VAriance (PERMANOVAs) were run to test for mineralogical differences between rock types. For multivariate data, constrained ordination Canonical Analysis of Principal coordinate (CAP) plots (*Anderson, Gorley & Clarke, 2008*) were used to visualise rock-related mineralogical differences. A leave-one-out procedure was used to test the allocation success of the discriminant function for rock groupings in CAP, with permutation tests used to test the significance of the trace test statistic and first canonical eigenvalue. Vector overlays of Spearman Rank correlations (for Rho values > 0.8) were used to identify the major minerals and trace elements that best characterised the mineralogy of each rock type.

## RESULTS

### Patterns of temperature difference on upper or lower surfaces of different rock types

Boulder upper and lower surfaces, for all six rock types, had generally a heterogeneous surface temperature differing in maxima and minima after four hours exposure to insolation on all days sampled (Fig. 1). Moving in any direction across boulder surfaces (a representative thermal image showing each rock type's surface temperature is provided in Fig. 2), temperature consisted of warmer and cooler areas with patch sizes <10 cm. When transects were drawn on thermal images, three qualitative patterns of temperature difference were identified. The first pattern was a temperature mosaic, which consisted of heterogeneous temperatures across the entire boulder surface (Fig. 3A). The temperature difference between the warmest and coolest mosaic areas was ≥5 °C. The second pattern was a temperature gradient, where temperature appeared to gradually decrease from the
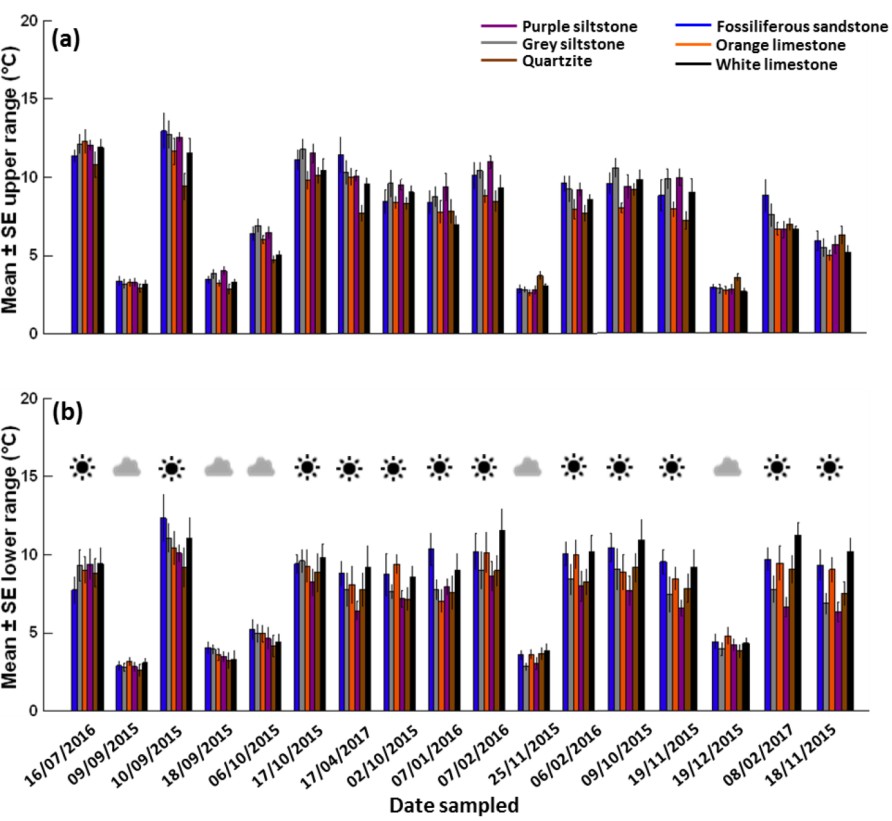

**Figure 1** **Mean ± SE temperature range ($n = 6$ boulders per rock type per day).** (A) Upper surfaces and (B) lower surfaces on different days (ordered by the daily maximum air temperature during sampling) during the common garden experiment. The sun or cloud symbols in panel b denote the day condition for each date sampled.

side nearest the sun to the side opposite (Fig. 3B). The temperature difference between the warmest and coolest gradient areas was ≥5 °C. The third pattern was limited temperature heterogeneity, which consisted of only small temperature differences <5 °C between the warmest and coolest areas (Fig. 3C).

The frequency of occurrence for these three patterns of temperature difference was strongly influenced by daily weather conditions (Table 2). On cloudy days, almost all (>94%) boulder surfaces for all six rock types were categorised as having limited temperature heterogeneity (Table 2). Temperature mosaics and gradients were seldom, if ever, observed on cloudy days (Table 2). On the two cooler sunny days, almost all (>94%) boulder surfaces for all six rock types were categorised as having either temperature gradients or mosaics (Table 2). Temperature gradients were more common than temperature mosaics, in a ratio of 3:1, on these sunny days (Table 2). On the hottest sunny day, all three patterns of temperature difference were observed (Table 2); however, boulder surfaces became generally hot, with temperature differences between the warmest and coolest areas mostly <5 °C (Table 2, Fig. 1). As a result, limited temperature heterogeneity was the most common (≥78%) temperature pattern identified on the hottest

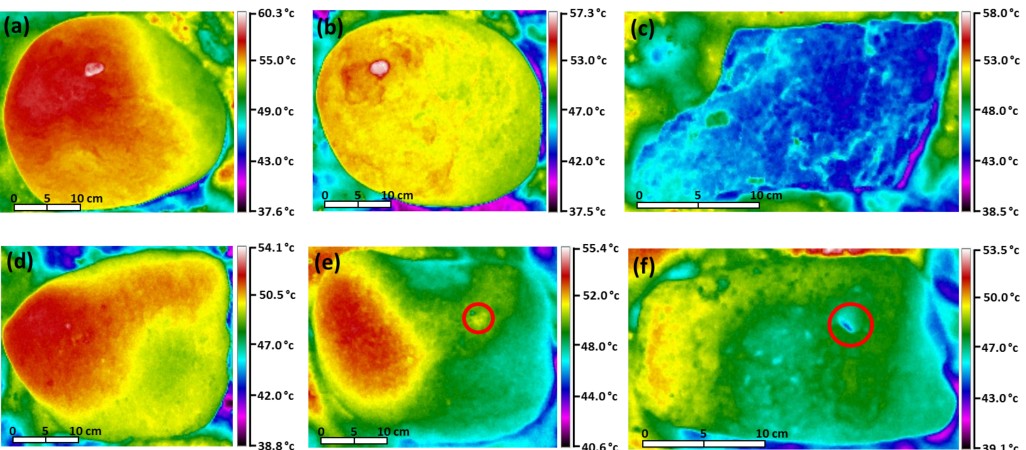

**Figure 2 Thermal images showing patterns of temperature difference on the upper surfaces of boulders.** (A) Grey siltstone, (B) purple siltstone, (C) quartzite, (D) fossiliferous sandstone, (E) orange limestone and (F) white limestone. Each thermal image was recorded after boulders were exposed to insolation for four hours on a sunny day (air temperature = 39 °C). Temperature scales (at right of each image) are specific to each rock type. The red circles in panels (E) and (F) depict examples of the interruption of gradient mosaics by microhabitat features.

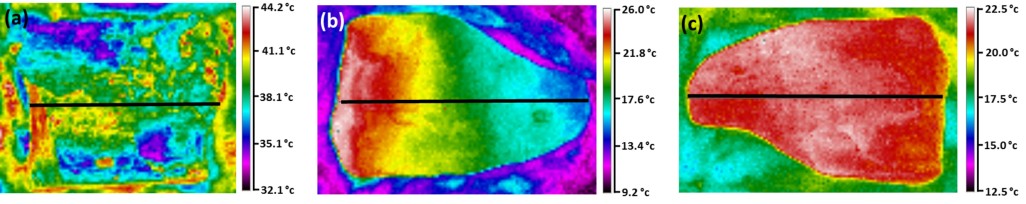

**Figure 3 Thermal images showing patterns of temperature difference on boulder surfaces.** (A) Mosaic on the upper surface of quartzite; (B) gradient on the upper surface of grey siltstone; and (C) limited heterogeneity on the upper surface of grey siltstone. Temperature scales (at right of each image) are specific to each image. The black horizontal line denotes the transect drawn on each image to quantify temperature patterns. The sun is on the left of each image.

sunny day (Table 2), with gradients and mosaics observed at much lower frequencies than on the other sunny days.

On sunny days patterns of temperature difference differed among rock types. The coarse, angular surfaces of quartzite had a mosaic of fine millimetre to centimetre scale patches of heterogeneous temperature (Fig. 2C, Appendix Table A1). The five remaining rock types all generally had temperature gradients, although the spatial arrangement of these gradients differed. The generally smooth and featureless surfaces of siltstone had simple gradients of warmer through cooler areas (Figs. 2A–2B, Appendix Table A1). In contrast, the two limestones and fossiliferous sandstone had complex surfaces intersected by shallow depressions and pits (<1 cm depth). Consequently, their temperature gradients were interrupted intermittently by these depressions and pits, which could be either warmer or cooler (by up to 2−3 °C) than the flatter surfaces immediately around them (Figs. 2D–2F).

**Table 2** **Frequencies of occurrence (%), pooled across rock types, for the three patterns of temperature difference and the orientation of the maximum in relation to the sun.** For boulder upper and lower surfaces ($n = 36$) for a subset of the cloudy and sunny days sampled.

| Weather | Date | Maximum air temperature (°C) | Surface | Temperature pattern (%) | | | Orientation (%) | |
|---|---|---|---|---|---|---|---|---|
| | | | | Mosaic | Gradient | Limited heterogeneity | Side facing sun | Any other side |
| Cloudy | 09/09/2015 | 15 | Upper | 0 | 0 | 100 | 91.6 | 8.4 |
| | | | Lower | 0 | 0 | 100 | 100 | 0 |
| | 25/11/2015 | 30 | Upper | 0 | 0 | 100 | 100 | 0 |
| | | | Lower | 0 | 0 | 100 | 100 | 0 |
| | 19/12/2015 | 38 | Upper | 0 | 0 | 100 | 97.2 | 2.8 |
| | | | Lower | 0 | 5.6 | 94.4 | 94.4 | 5.6 |
| Sunny | 16/07/2016 | 12 | Upper | 13.9 | 86.1 | 0 | 100 | 0 |
| | | | Lower | 13.9 | 80.5 | 5.6 | 100 | 0 |
| | 07/02/2016 | 29 | Upper | 22.2 | 75.0 | 2.8 | 100 | 0 |
| | | | Lower | 25.0 | 75.0 | 0 | 94.4 | 5.6 |
| | 18/11/2015 | 40 | Upper | 2.8 | 19.4 | 77.8 | 100 | 0 |
| | | | Lower | 5.6 | 33.3 | 61.1 | 94.4 | 5.6 |

All three patterns of temperature difference were related to boulder orientation, with the hottest temperatures generally recorded for the side of boulders nearest the sun (>92%, Table 2). Each rock type generally had the same pattern of temperature difference on its upper and lower surfaces.

## Maximum temperature differs between rock types and surfaces

After four hours exposure the mean maximum temperature was hotter than the air temperature for all six rock types on both surfaces, especially on sunny days (Figs. 4A–4B). The maximum temperature of upper and lower surfaces generally increased with time, with maxima often peaking around four hours and plateauing thereafter (Fig. 5A). The largest increases in maxima were generally recorded during the first two hours exposure to insolation, with smaller increases (and sometimes decreases) recorded thereafter (Appendix Figs. A3 & A4). After four hours, the hottest maximum recorded was 57.8 °C (sunny day, air temperature = 39 °C) for the upper surface of grey siltstone while the coolest maximum was 14.4 °C (sunny day, air temperature = 12 °C) for the lower surface of quartzite. Over four hours, increases in mean maximum surface temperature >20 °C were recorded for some rock types on several days, with the greatest increases recorded for upper surfaces on sunny days (Appendix Fig. A5).

When rock types were ranked from hottest to coolest mean maximum temperature after four hours, a consistent rank order across replicate days was identified, irrespective of weather conditions (Table 3). For both surfaces, the two siltstones consistently recorded the hottest maxima, followed by fossiliferous sandstone and orange limestone in descending rank order (Table 3, Fig. 5A). On upper surfaces, white limestone consistently recorded the second coolest maxima and quartzite the coolest, while on lower surfaces both rock types were equally ranked in terms of the coolest maxima (Table 3). After four hours exposure for the 17 replicate days, the smallest difference for mean maxima across the six rock types
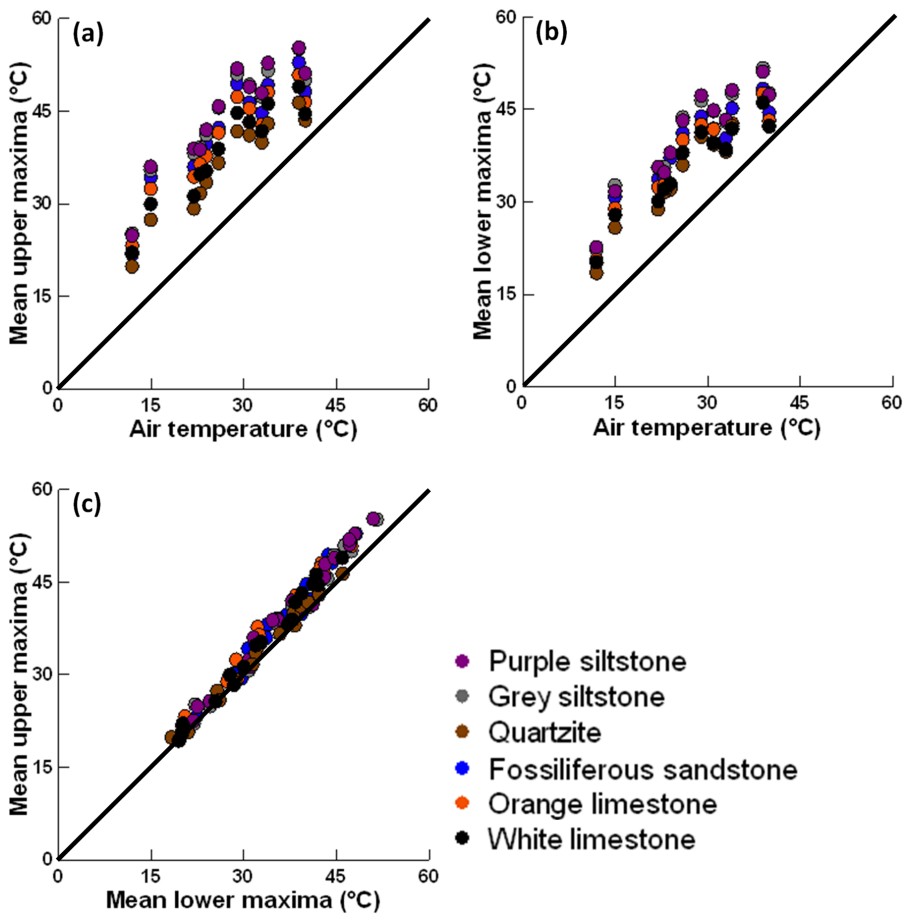

**Figure 4** Scatter plots (1:1 line shown) showing the association between (A) air temperatures and mean upper maxima; (B) air temperatures and mean lower maxima; and (C) mean upper maxima and mean lower maxima after four hours for each rock type on each sunny day.

was 2.5 °C, while the largest difference was 10.2 °C. A similar rank order was identified for the change in maxima over four hours for both surfaces (Table 3, Appendix Fig. A5). The two siltstones generally had the largest increase in maximum temperature, while white limestone and quartzite had the smallest (Table 3, Appendix Fig. A5).

Generally, maxima behaved similarly on upper and lower surfaces (Fig. 4C). At the commencement of sampling on most days, small negative differences were detected between upper and lower surface maxima for all rock types, with lower surfaces having mean maxima that were slightly hotter (<2 °C) than upper surfaces (Appendix Fig. A6). Thereafter, small positive differences were detected between upper and lower surfaces for most rock types, with upper surfaces having hotter mean maxima than lower surfaces, although these differences never exceeded 5 °C (Fig. 5B, Appendix Fig. A6). The only notable exception to this trend was quartzite, which generally had small negative differences throughout, with lower surfaces sometimes having hotter mean maxima than upper surfaces (Fig. 5B). The difference in maxima between upper and lower surfaces was always smallest on cloudy

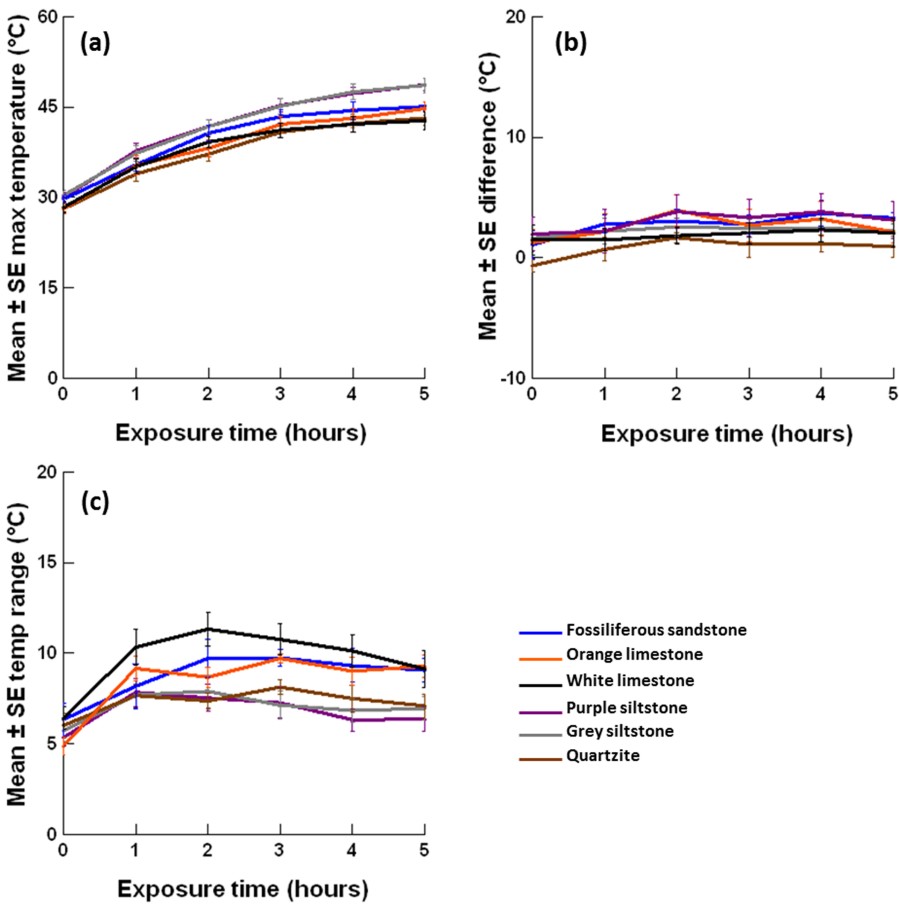

**Figure 5** Mean ± SE (A) lower-surface maximum temperature; (B) maxima difference between upper and lower surfaces and (C) lower-surface temperature range for 6 rock types ($n = 6$ per rock type) over five hours of exposure to insolation on November 18, 2015. Each $y$-axis extends to encompass the range of raw data. Please see the appendices for additional figures for the upper surfaces and other days sampled.

days, with larger differences detected on sunny days (Appendix Fig. A6). Minima behaved similarly to maxima over four hours exposure to insolation with the same trends identified for rock type, surface and time exposed (refer to minima sub-section in Appendix for more details).

## Temperature range does not consistently differ between rock types or surfaces

Generally, temperature range behaved similarly on upper and lower surfaces (Appendix Figs. A7 & A8). Temperature range was influenced by weather conditions (Fig. 1). A larger temperature range (5–15 °C) that was more variable between rock types and exposure times was recorded on sunny days for both surfaces (Fig. 5C, Appendix Figs. A7 & A8). In contrast, a smaller temperature range (generally <5 °C) that was less variable between rock types and exposure times was recorded on each cloudy day for both surfaces (Appendix Figs. A7 & A8). After four hours, the largest temperature range recorded was 16.3 °C (sunny

Janetzki et al. (2021), *PeerJ*, DOI 10.7717/peerj.10712

**Table 3** **The daily rank order of rock types from largest to smallest (6 = largest, 1 = smallest) maximum temperature after four hours and change in maximum temperature over four hours for upper and lower surfaces.** Entries are the cumulative number of occurrences of that daily rank for each rock type. The rank sum (sum of daily ranks) was used to assign an overall rank to each rock type from largest to smallest (6 = largest value, 1 = smallest).

| Surface Daily rank | | | | | | | | Upper | | | | | | | | | Lower | |
|---|---|---|---|---|---|---|---|---|---|---|---|---|---|---|---|---|---|---|
| Measure | Rock type | 6 | 5 | 4 | 3 | 2 | 1 | Rank sum | Overall Rock rank | 6 | 5 | 4 | 3 | 2 | 1 | Rank sum | Overall Rock rank |
| Maximum temperature after four hours | Purple siltstone | 14 | 3 | 0 | 0 | 0 | 0 | 99 | **6** | 7 | 10 | 0 | 0 | 0 | 0 | 92 | **5** |
| | Grey siltstone | 3 | 14 | 0 | 0 | 0 | 0 | 88 | **5** | 10 | 7 | 0 | 0 | 0 | 0 | 95 | **6** |
| | Fossiliferous sandstone | 0 | 0 | 16 | 1 | 0 | 0 | 67 | **4** | 0 | 0 | 16 | 1 | 0 | 0 | 67 | **4** |
| | Orange limestone | 0 | 0 | 1 | 16 | 0 | 0 | 52 | **3** | 0 | 0 | 1 | 15 | 1 | 0 | 51 | **3** |
| | White limestone | 0 | 0 | 0 | 0 | 13 | 4 | 30 | **2** | 0 | 0 | 0 | 0 | 9 | 8 | 26 | **1** |
| | Quartzite | 0 | 0 | 0 | 0 | 4 | 13 | 21 | **1** | 0 | 0 | 0 | 1 | 7 | 9 | 26 | **1** |
| Change in maximum temperature over four hours | Purple siltstone | 8 | 7 | 1 | 0 | 1 | 0 | 89 | **6** | 9 | 3 | 4 | 0 | 0 | 1 | 86 | **5** |
| | Grey siltstone | 7 | 8 | 0 | 1 | 1 | 0 | 87 | **5** | 5 | 10 | 0 | 2 | 0 | 0 | 86 | **5** |
| | Fossiliferous sandstone | 2 | 0 | 14 | 1 | 0 | 0 | 71 | **4** | 1 | 4 | 6 | 6 | 0 | 0 | 68 | **4** |
| | Orange limestone | 0 | 1 | 2 | 13 | 1 | 0 | 54 | **3** | 1 | 0 | 6 | 6 | 4 | 0 | 56 | **3** |
| | White limestone | 0 | 1 | 0 | 1 | 9 | 6 | 32 | **2** | 1 | 0 | 0 | 2 | 3 | 11 | 29 | **2** |
| | Quartzite | 0 | 0 | 0 | 1 | 5 | 11 | 24 | **1** | 0 | 0 | 1 | 1 | 10 | 5 | 32 | **1** |
day, air temperature = 15 °C) on the upper surface of grey siltstone while the smallest temperature range was 1.6 °C (cloudy day, air temperature = 38 °C) on the upper surface of purple siltstone.

When rock types were ranked from largest to smallest for the mean temperature range after four hours, there was little evidence of a consistent ranking across replicate days for both upper and lower surfaces (Table 4, Appendix Figs. A7 & A8). Rankings were similarly variable after both shorter and longer exposure times, with rank order often changing from one exposure time to the next (Appendix Figs. A7 & A8). No consistent ranking was identified either for the change in mean temperature range over four hours for either surface (Table 4, Appendix Fig. A9).

The temperature range difference between upper and lower surfaces (i.e., upper range–lower range) was always smallest on cloudy days, with larger differences detected on sunny days (Appendix Fig. A10). On cooler days with a maximum daily air temperature <30 °C, regardless of the day condition, small positive differences were generally detected, with the upper surfaces of most boulders having a larger temperature range than lower surfaces (Appendix Fig. A10). In contrast, on hotter days with a maximum daily air temperature ≥30 °C, small positive differences were detected only for the two siltstones. Small negative differences were often measured for the two limestones and quartzite, with lower surfaces having a larger temperature range than upper surfaces (Appendix Fig. A10). Overall, the two siltstones generally had the largest range difference between upper and lower surfaces, while white limestone and quartzite often had the smallest (Appendix Fig. A10).

## Rock-related differences in maximum temperature are correlated with their mineral composition

Silicon dioxide ($SiO_2$) and calcium oxide (CaO) were the dominant major minerals detected, with orange limestone having a CaO-dominated mineralogy and all other rock types a $SiO_2$-dominated mineralogy (Appendix Table A3). Major mineral composition significantly differed among rock types (PERMANOVA permuted $p$-value = 0.0001). For rock type differences, a CAP constrained-ordination plot used five axes to discriminate major-mineral differences, with the first two axes accounting for 99.54% (prop. G) of the total mineralogical variability (Fig. 6A). All samples were correctly classified using a leave-one-out procedure, and permutation tests for both the trace test statistic ($p = 0.0001$) and first canonical eigenvalue ($p = 0.0001$) were highly significant. The vector overlay of Spearman rank correlations (for rho values > 0.8) for major minerals associated with rock differences showed that each rock type had a specific major-mineral composition (Fig. 6A). Grey siltstone was characterised by higher aluminium oxide and potassium oxide contents, quartzite by the highest $SiO_2$ content, and orange limestone and fossiliferous sandstone by higher CaO contents (Fig. 6A). Rock type differences in the content of specific major minerals were also detected for 10 from 11 major minerals (largest significant permuted PERMANOVA $p$-value = 0.0330 for magnesium oxide), with only sulfur trioxide not differing between rock types (PERMANOVA permuted $p$-value = 0.1244).

Trace-element composition significantly differed among rock types (PERMANOVA permuted $p$-value = 0.0001). For rock type differences, a CAP constrained-ordination plot

Janetzki et al. (2021), *PeerJ*, DOI 10.7717/peerj.10712

**Table 4  The daily rank order of rock types from largest to smallest (6 = largest value, 1 = smallest) temperature range after four hours and change in temperature range over four hours for upper and lower surfaces.** Entries are the cumulative number of occurrences of that daily rank for each rock type. The rank sum (sum of daily ranks) was used to assign an overall rank to each rock type from largest to smallest (6 = largest value, 1 = smallest).

| Surface Daily rank | | | | | | | | | | Upper | | | | | | | | Lower | |
|---|---|---|---|---|---|---|---|---|---|---|---|---|---|---|---|---|---|---|---|
| Measure | Rock type | 6 | 5 | 4 | 3 | 2 | 1 | Rank sum | Overall rock rank | 6 | 5 | 4 | 3 | 2 | 1 | Rank sum | Overall rock rank |
| Temperature range after four hours | Purple siltstone | 4 | 4 | 5 | 2 | 1 | 1 | 73 | **4** | 0 | 1 | 1 | 4 | 3 | 8 | 35 | **1** |
| | Grey siltstone | 4 | 9 | 1 | 3 | 0 | 0 | 82 | **6** | 0 | 3 | 2 | 6 | 5 | 1 | 52 | **3** |
| | Fossiliferous sandstone | 5 | 2 | 7 | 2 | 1 | 0 | 76 | **5** | 5 | 8 | 2 | 1 | 0 | 1 | 82 | **5** |
| | Orange limestone | 1 | 0 | 1 | 5 | 6 | 4 | 41 | **2** | 3 | 0 | 9 | 3 | 1 | 1 | 66 | **4** |
| | White limestone | 0 | 2 | 2 | 4 | 7 | 2 | 46 | **3** | 9 | 4 | 2 | 0 | 2 | 0 | 86 | **6** |
| | Quartzite | 3 | 0 | 1 | 1 | 2 | 10 | 39 | **1** | 0 | 1 | 1 | 3 | 6 | 6 | 36 | **2** |
| Change in temperature range over four hours | Purple siltstone | 3 | 4 | 3 | 4 | 2 | 1 | 67 | **5** | 1 | 0 | 1 | 5 | 5 | 5 | 40 | **1** |
| | Grey siltstone | 7 | 3 | 3 | 2 | 1 | 1 | 78 | **6** | 0 | 4 | 2 | 1 | 8 | 2 | 49 | **3** |
| | Fossiliferous sandstone | 2 | 5 | 3 | 2 | 2 | 3 | 62 | **4** | 4 | 8 | 3 | 0 | 0 | 1 | 77 | **6** |
| | Orange limestone | 1 | 1 | 4 | 4 | 4 | 3 | 50 | **2** | 7 | 2 | 4 | 1 | 0 | 3 | 74 | **5** |
| | White limestone | 3 | 2 | 3 | 1 | 6 | 2 | 57 | **3** | 5 | 3 | 4 | 2 | 2 | 1 | 72 | **4** |
| | Quartzite | 1 | 2 | 1 | 4 | 2 | 7 | 43 | **1** | 0 | 0 | 2 | 8 | 2 | 5 | 41 | **2** |

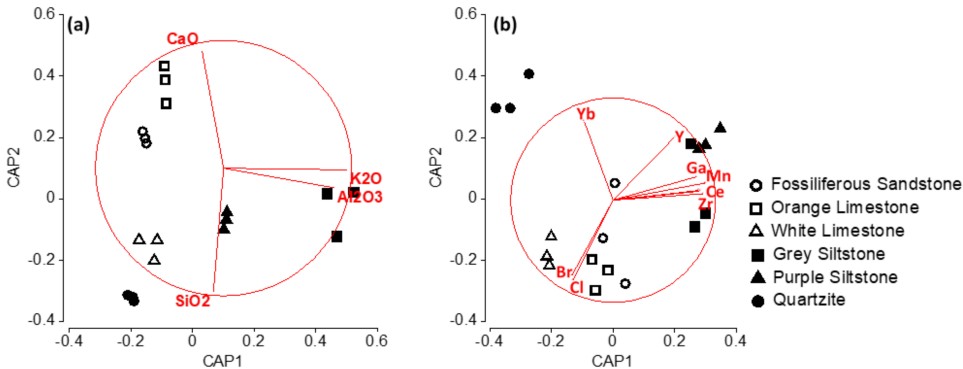

**Figure 6 Constrained ordination CAP plots with vector overlay of Spearman rank correlations (for rho values > 0.8) for (A) major minerals and (B) trace elements contributing to differences in mineralogy among rock types.** Each point represents a single rock sample tested. CaO = calcium oxide; $SiO_2$ = silicon dioxide; $K_2O$ = potassium oxide; $Al_2O_3$ = aluminium oxide; Yb, ytterbium; Y, Yttrium; Br, bromine; Cl, chlorine; Zr, zirconium; Ce, cerium; Mn, manganese and Ga, gallium.

used two axes to discriminate trace-element differences, with these two axes accounting for 79.8% (prop. G) of the total mineralogical variability (Fig. 6B). Some 88.9% of samples were correctly classified using a leave-one-out procedure, and permutation tests for both the trace test statistic ($p = 0.0001$) and first canonical eigenvalue ($p = 0.0001$) were highly significant. The vector overlay of Spearman rank correlations (for rho values > 0.8) for trace elements associated with rock type differences suggested that each rock type had a specific trace-element composition (Fig. 6B). The two siltstones were characterised by a higher trace-metal content (manganese and zirconium especially), quartzite by generally low trace-element quantities (although it had the highest ytterbium content), and the two limestones and fossiliferous sandstone by a higher chlorine content (Fig. 6B, Appendix Table A4).

The vector overlay of Spearman rank correlations (for rho values > 0.8) in CAP plots (Fig. 6) showed that the two siltstones, which consistently had the hottest maxima, had mineralogies that were characterised by a higher content of metallic oxides and trace metals versus all other rock types (Fig. 6, Appendix Tables A3 & A4). White limestone and quartzite, which consistently had the coolest maxima, were characterised by the highest content of $SiO_2$ and the lowest content of most metallic oxides and trace metals versus all other rock types (Fig. 6, Appendix Tables A3 & A4). Meanwhile, orange limestone and fossiliferous sandstone, which had intermediate maximum temperatures, were characterised by higher contents of CaO and chlorine, and metallic oxide and trace metal quantities that were generally lower than the two siltstones but greater than white limestone and quartzite (Fig. 6, Appendix Tables A3 & A4).

## DISCUSSION

Under common-garden conditions that simulated low tide, we were able to isolate temperature that was a function of the rocks themselves, not their setting. Weather

condition was the largest determinant of boulder temperature, with cloud cover moderating all temperature dependent variables. Air temperature was strongly associated with boulder temperature, with the hottest boulder temperatures measured on the hottest sunny days. Upper and lower surfaces had patterns of temperature difference, with three qualitative patterns identified: gradients; mosaics; and limited heterogeneity. On cloudy days, limited heterogeneity was identified on most surfaces for all rock types. On the hottest sunny day sampled, and with rocks greatly heated, limited heterogeneity was again the dominant pattern identified, although some temperature gradients and mosaics were observed. On the remaining sunny days, most quartzite surfaces developed temperature mosaics, while the surfaces of all other rock types generally developed temperature gradients. The maximum (and minimum) temperature differed consistently between rock types and surfaces. Upper surface maxima were generally hotter (<5 °C) than lower surface maxima, with the two siltstones consistently being the hottest and quartzite and white limestone the coolest. Each rock type had a unique mineralogy. The maximum temperature correlated with the metallic oxide and trace metal content of rocks, with the hottest rock types having the highest metallic oxide and trace metal contents.

These results provide evidence that rock type is likely to contribute to the heterogenous patterns of temperature observed across rocky seashores. Under common-garden conditions, differences in maximum surface temperature of up to 10.2 °C were identified among the six rock types. This experimental observation supports field data previously collected on rocky seashores, which show different rock types can have different surface temperatures (*Raimondi, 1988*; *Marshall, McQuaid & Williams, 2010*; *Judge, Botton & Hamilton, 2011*). Besides differences in temperature, the six rock types investigated also support different patterns of temperature variation across the boulder surface. For example, temperature mosaics were generally observed on quartzite boulders only. These temperature mosaics may have developed because of the specific microtopographic features of quartzite (i.e., coarse, angular surfaces), with microtopography associated with temperature heterogeneity on rocky seashores elsewhere (*Lathlean, Ayre & Minchinton, 2012*; *Choi et al., 2019*). Alternatively, these mosaics may reflect differences in long-wave emissivity not accounted for by our use of a generic emissivity value. Future studies should investigate emissivity variation across quartzite surfaces to provide a better understanding of how microtopography and emissivity interact to create temperature mosaics. Microhabitat features like cracks and depressions were unique to limestone and sandstone boulders here. These microhabitat features were always a different temperature to the flat surfaces immediately around them, with microhabitat features associated with temperature heterogeneity on other rocky substrates (*Chapperon & Seuront, 2011*; *Judge, Botton & Hamilton, 2011*; *Lathlean et al., 2015*). Extrapolating these results to the larger spatial scale of natural rocky seashores, our experimental results indicate that rock type contributes to the temperature heterogeneity observed within and among rocky seashores.

Patterns of temperature difference were related to boulder orientation, with the hottest locations on all rocks generally observed on the side of boulders nearest the sun. Substrate orientation appears to affect the spatial arrangement of temperatures on rocky seashores generally, with rock faces orientated towards the sun (*Harley, 2008*; *Seabra et al., 2011*

*Chapperon et al., 2016*; *Chapperon, Studerus & Clavier, 2017*; *Aguilera, Arias & Manzur, 2019*) having the hottest surface temperatures. The potential biological relevance of the temperature patterns on boulder surfaces identified in this study have not yet been quantified. However, given that periwinkles will select cooler locations when offered centimetre-scale temperature gradients (*Soto & Bozinovic, 1998*), and intertidal ectotherms generally can respond to habitat-scale temperature mosaics (e.g., *Garrity, 1984*; *Chapperon & Seuront, 2011*; *Judge, Botton & Hamilton, 2011*; *Chapperon, Le Bris & Seuront, 2013*), it is possible that mobile intertidal ectotherms may respond to the temperature patterns on boulder surfaces described here.

On the hottest sunny day sampled, boulder surfaces became generally uniformly hot with limited temperature heterogeneity. These uniformly hot boulder surfaces may represent the upper threshold of extreme thermal habitats for intertidal biota on boulders with a maximum dimension <30 cm, as the boulders sampled here were relatively thin. Given the risks that desiccation and heat stress pose to organism survival and fitness (e.g., *Jones & Boulding, 1999*; *Harley, 2008*; *Gedan et al., 2011*; *Monaco et al., 2015*), and the exacerbation of these risks at the hottest environmental temperatures (*Harley, 2008*; *Seuront et al., 2019*), the disappearance of temperature mosaics and gradients on hot, sunny days may be problematic for intertidal biota. If the cooler areas of mosaics or gradients are found to function as thermal refuges for intertidal biota, but these refuges disappear on the hottest days when they are needed most, then the thermal quality of these boulder habitats may be diminished on hot, sunny days. Consequently, organism survival and fitness may be challenged.

Maximum temperatures differed consistently among rock types, with the two siltstones always hottest and white limestone and quartzite the coolest. Rock types with the coolest temperatures were also the most thermally stable, as they had the smallest temperature increases while exposed to insolation over four hours. Thus, some rock types possibly minimise thermal stress to biota more than others. The mechanisms that resulted in some rock types warming more slowly than others were not investigated here, but may be related to rock type differences in the amount infrared energy absorbed/reflected or the thermal conductance of specific mineral constituents (*Seuront, Ng & Lathlean, 2018*). Variations in thickness also affect the thermal stability of boulders (*Huey et al., 1989*), but as the boulders sampled here were standardised for size and were relatively thin, variations in thickness are unlikely to account for the different warming rates of these six rock types.

The thermal benefits of cooler rock types may be observed across the entire vertical gradient of the rocky intertidal zone. Lower on the shore, where some biota have only low thermal tolerances (*McMahon, 1990*; *Madeira et al., 2012*), cooler rock types that only warm small amounts when emersed may offer thermal refuges to thermally-sensitive species. Higher up the shore, where biota generally have greater thermal tolerances (*McMahon, 1990*; *Madeira et al., 2012*), cooler rock types may not warm to potentially deleterious temperatures during extended periods of emersion. These characteristics have not been recognised to date but could allow some predictions of the future fate of populations and assemblages on rocky seashores.

Cooler rock temperatures have been positively associated with intertidal biota on seashores globally. In Mexico, the higher vertical distribution of barnacles on granite than basalt shores was attributed to granite's cooler surface temperatures (*Raimondi, 1988*), while in Brunei Darussalam snail mimics had cooler body temperatures on lighter-coloured sandstone than darker-coloured ferruginous sandstone (*Marshall, McQuaid & Williams, 2010*). In Australia, barnacle recruitment and growth rate was higher on cooler than hotter areas of grey siltstone platform (*Lathlean, Ayre & Minchinton, 2013*), while in Brazil and Panama, post-settlement mortality of barnacles was higher on hotter black plates than cooler white plates (*Leal et al., 2020*). Moreover, invertebrate and algal abundance and richness was negatively related with peaks in substrate temperature in Chile (*Aguilera, Arias & Manzur, 2019*), while on igneous seashores in Panama, gastropod body temperatures and mortality were highest in areas with the hottest temperature (*Garrity, 1984*). Therefore, the cooler and more thermally-stable rock types identified in this study such as white limestone and quartzite may function as thermal refuges for some intertidal biota. If boulders comprised of cooler rock types are available to biota seeking refuge, then the thermal benefits that cooler rocks potentially confer may improve the chances of biota surviving while emersed, and thus persisting, compared to hotter rock types such as siltstone.

In this study, the metallic oxide and trace metal content of rocks correlated positively with higher maximum temperatures. On the basis of these results, metallic minerals may increase the thermal conductance of some rock types and increase the amount of infrared radiation they absorb (*Seuront, Ng & Lathlean, 2018*). However, mineralogy is just one potential driver of the temperature differences identified here. Differences in colour (*Raimondi, 1988*; *Marshall, McQuaid & Williams, 2010*; *Judge, Botton & Hamilton, 2011*) and microtopography (*Lathlean, Ayre & Minchinton, 2012*; *Choi et al., 2019*) have been shown previously to affect rock temperature, with the six rock types investigated here also differing in these attributes. It is possible that the smooth, featureless surfaces of siltstone were able to reach hotter surface temperatures versus the coarse, textured surfaces of limestone and quartzite. Moreover, darker rocks of the same type (i.e., purple versus grey siltstone and orange versus white limestone, Appendix Fig. A2) attained hotter surface temperatures. However, as colour and microtopography differences between rock types were not quantitatively measured it is difficult to reliably associate either variable with any rock-related temperature differences.

Lower surfaces generally had cooler maximum temperatures ($<5\,°C$) than upper surfaces. Over four hours exposure to insolation, a greater increase in maxima was measured on boulder upper than lower surfaces. Therefore, lower surfaces have some temperature benefits over upper surfaces due to their shaded surfaces generally providing cooler maxima and slower temperature increases. The thermal benefits of living underneath boulders are often cited (*Evans, 1948*; *Chapperon & Seuront, 2011*; *Chapperon, Le Bris & Seuront, 2013*). However, our results, in conjunction with those published previously for the same siltstone boulders (*Chapperon & Seuront, 2011*; *Chapperon, Le Bris & Seuront, 2013*), suggest that the magnitude of temperature relief that biota experience under boulders on the mid-lower seashore at this site may be quite small ($<5\,°C$) when compared to temperature differences

of up to 25.5 °C between sun-exposed and sun-protected microhabitats elsewhere on rocky seashores (e.g., *Denny et al., 2011*; *Lathlean, Ayre & Minchinton, 2012*). These small temperature differences between upper and lower surfaces may be attributed to the relatively thin boulders that naturally occur on the Fleurieu Peninsula, with thicker or larger boulders on seashores elsewhere potentially having larger temperature differences between upper and lower surfaces. Nevertheless, from a physiological perspective, this 5 °C difference between the top and bottom of boulders may help to ensure organisms remain within their thermal tolerance limits (*Helmuth et al., 2002*). Moreover, the combined benefits of cooler surface temperatures plus under-boulder dampness and shading from insolation (*Evans, 1948*; *Chapman, 2003*; *Chapperon, Le Bris & Seuront, 2013*) may interact to make lower surfaces a thermally favourable habitat for intertidal biota at low tide.

Temperature range was highly variable among the six rock types, with no single rock type having a temperature range that was consistently distinct from the others. If these sorts of results extend to the seashore, then biota would have ample opportunity to respond to the range of temperatures on all rock types, as no single rock type had a temperature range larger or smaller than the others. In this study, a temperature range as large as 16.3 °C was recorded across an individual boulder surface. This temperature range is larger than the 5.0 °C (*Leal et al., 2020*) or 8.2 °C (*Lathlean, Ayre & Minchinton, 2012*) maximum ranges recorded among replicate quadrats sampled on the same rock platforms, but considerably smaller than the 24.0 °C maximum range detected between between the edge and centre of rocks (maximum length <2 m) sheltering garter snakes (*Huey et al., 1989*), the 25 °C maximum range detected between replicate boulders sampled on the same seashore (*Gunderson et al., 2019*), or the 25.5 °C maximum range detected between different seashore microhabitats (*Chapperon & Seuront, 2011*; *Chapperon, Studerus & Clavier, 2017*). Consequently, these new results for single boulder surfaces indicate that temperature range is likely be specific to the type(s) of substrate, habitat and region investigated.

## CONCLUSIONS

Under common-garden conditions, the upper and lower surfaces of boulders had patterns of temperature difference, with three qualitative patterns identified: gradients; mosaics; and limited heterogeneity. Maximum temperature differed consistently between rock types and surfaces, with upper boulder surfaces generally <5 °C warmer than lower boulder surfaces. Each rock type had a unique major mineral and trace elemental composition, with the content of metallic oxides and trace metals in rock types correlating with their maximum temperature. The lower surface of rock types with the lowest metallic oxide and trace metal content (quartzite and white limestone in this study) potentially offer the best thermal refugia for intertidal biota on the mid-lower seashore during summer. Consequently, these results show that rock type, mineralogy, and boulder surface, which have largely been neglected from investigations of substrate temperature until now, can play important roles in contributing to the temperature heterogeneity observed on rocky seashores globally.

## ACKNOWLEDGEMENTS

This work is dedicated to the memory of our mentor and co-author Peter G. Fairweather, who sadly passed away prior to the publication of this work. He was an incredible marine ecologist, and an even better friend. He had a profound and lasting impact on every scientist he inspired, ourselves included. We are grateful to S. Hawkins, P. Raimondi, B. Helmuth and two anonymous referees for useful comments and suggestions on earlier versions of this work. We are also grateful to C. Flaxman for identifying rock samples and to M. Raven (CSIRO Land and Water, Adelaide) for completing and commenting upon XRF analyses of rock samples.

### Funding

The research in this manuscript was funded by an Australian Government Research Training Program Scholarship (NJ) and by the Holsworth Wildlife Research Endowment –Equity Trustees Charitable Foundation. The funders had no role in study design, data collection and analysis, decision to publish, or preparation of the manuscript.

### Grant Disclosures

The following grant information was disclosed by the authors:
Australian Government Research Training Program Scholarship (NJ).
Holsworth Wildlife Research Endowment—Equity Trustees Charitable Foundation.

### Competing Interests

The authors declare there are no competing interests.

### Author Contributions

- Nathan Janetzki conceived and designed the experiments, performed the experiments, analyzed the data, prepared figures and/or tables, authored or reviewed drafts of the paper, and approved the final draft.
- Kirsten Benkendorff conceived and designed the experiments, prepared figures and/or tables, authored or reviewed drafts of the paper, and approved the final draft.
- Peter G. Fairweather conceived and designed the experiments, prepared figures and/or tables, authored or reviewed drafts of the paper, provided materials and experimental site, and approved the final draft.

### Field Study Permissions

The following information was supplied relating to field study approvals (i.e., approving body and any reference numbers):

Dr Gillian Napier provided access to her farm, thus enabling us to construct the boulder plot where this common-garden experiment was run.

### Data Availability

The thermal image raw data are available in a Supplemental File.

## Supplemental Information

Supplemental information for this article can be found online at http://dx.doi.org/10.7717/peerj.10712#supplemental-information.

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
