# Peer review of "Rocks of different mineralogy show different temperature characteristics: implications for biodiversity on rocky seashores"

_PeerJ, doi:10.7717/peerj.10712_

## Round 0.1 · original submission · Major Revisions

All referees compliment the presentation of the paper. One referee recommends rejection because the rationale behind the study is not clear. Can you answer the "so what?" question following this work - why is it useful? A second only gave a light critique without going into the depth regarding the study's purpose and how it advances knowledge. The third referee's comments align with the first but they recommended major revision and make several good suggestions, including being more concise and focused. Thus, because the paper needs significant thought with regard to explaining its rationale and how it advances scientific understanding, being much more focused and concise, I recommend Major Revision and further review before a decision can be reached on its suitability for PeerJ.

Reviewer 1 ·

Basic reporting

Clear and unambiguous, professional English is used throughout. Literature references and sufficient field background/context are provided. Tables and figures are appropriate. Hypotheses are not stated - objectives refer to methodology.

Experimental design

The question is well defined but not meaningful. Too what extent does a boulder on turf mirror rocky shore conditions?

Validity of the findings

Impact under real circumslances on the shore was not assessed.

Additional comments

This study involved collecting temperature data from the surfaces of rocks, with a view to better understanding the thermal heterogeneity experienced by intertidal rocky-shore life. The idea was (I think) to deconstruct this typically complex set of circumstances that influence body temperatures, by isolating rocks and studying them away from the shoreline. Essentially, the study aimed to describe how rock type, elemental composition, and upper or lower surface affected the temperature. The results were always very predictable; for example the upper surfaces become hotter than lower surfaces, and so on.

I am not convinced that the idea behind this project has really worked out, and that the findings contribute to improving understanding of rocky-shore thermal heterogeneity. It is unclear how these data can be applied or used to predict the temperature conditions on a shore. In my view there is no clear context for the data, and perhaps some effort should have been made to integrate the data into a predictive heat budget model. I also had several issues with the methods; too what extent does a boulder on turf mirror rocky shore conditions?

Whereas rock substratum characteristics are important there are several other equally important factors that effect thermal regimes on rocky shores, such as angle of the shoreline, gradient of the surface (azimuth), wind, and so on. All of these conditions vary from one shore to the next, and some vary within a shore at at fine spatialscale. All should be built into a heat budget model to predict shoreline temperatures.

Reviewer 2 ·

Basic reporting

No comment

Experimental design

no comment - see below

Validity of the findings

no comment - see below

Additional comments

Review of article 50962v1
Janetzki et al
General comments

Abstract
Fine no comments
Introduction
L7: Probably not necessary as this is common knowledge…..or should be.
L98: talks of ‘behaviour’ but that is not really a feature of inanimate objects, would suggest changing to ‘thermal regime’.
L101-106: These objectives are confusing. Not sure how objectives 1, 2, and 3 differ? Also if you are looking at maxima, wouldn’t you also specify minima? Objective 4, I would have thought this was implicit in the fact that they are different rock types and this would have been quantified before experiment began, thus not really a specific objective.
Methods
L114: what volume were the boulders? Also was shape standardized? Ideally they would be spherical or as close as possible to ensure comparisons were accurate and fair. A long sausage shaped boulder will not have the same thermal properties as a squat spherical one. More detail around this is needed.
Results:
Fine, potentially a bit long.
Discussion
L528: need to distinguish that this may be more problematic for sessile organisms such as oyster’s mussels etc. Mobile invertebrates will either seek boulder undersides or enter heat coma and roll down shore.
Importantly however, need to make mention that a significant caveat is differing rock dimensions tested….this is highly important for thermal properties of what you are testing and may confound some of your findings.

·

Basic reporting

This paper is rather lengthy for the amount of new material presented, and there are several places where more detail and clarification are needed (see below in general comments). Overall however the paper is clearly written.

Experimental design

There are some issues with the issue of the infrared camera using factory settings (see below).

Validity of the findings

See below.

Additional comments

X et al. experimentally explore the role that role minerology may play in driving temperature patterns in rocky intertidal shores during aerial exposure at low tide. I agree wholeheartedly with the authors that this is an understudied but potentially important mechanism that to this point has been largely ignored. This study provides an interesting entre into this question.

The introduction, while well-written, really needs to take a deeper dive into the mechanisms of why rock characteristics might be expected to drive patterns of temperature, and how this would be expected to influence patterns in the field. Doing so would also clear up some mis-statements that are made. For example, in several places the authors focus on the top sides vs undersides of rocks, and are correct that a major driver of differences is simply sun vs shade. But, as Ray “hot rocks” Huey et al. first showed in 1989, this is also strongly affected by the size (thickness) of the rock, which belies the idea that one can simply compare “tops vs bottoms” as in line 31.

In brief, there are several physical reasons why rock physical characteristics can drive temperatures, and these really need to be considered more carefully when interpreting these results and recognizing potential limitations of the approaches used. First, rock surface characteristics (albedo/reflectivity) can affect the total amount of solar short-wave radiative energy that is absorbed, i.e. white chalky rocks likely reflect more solar energy than do dark basaltic rocks. Similarly, the surface emissivity of the rock determines release of infrared (long wave) energy. This is important, because the team measured surface temperatures using an infrared camera set at the factory default of 0.95 (line 187). This is probably ok, but it is important to note that because this wasn’t tuned to each rock type (which would be done by changing the emissivity on the camera until it recorded a temperature that matched that of a thermocouple) any potential role of emissivity is ignored. Once the heat energy enters the rock, the temperature change will be largely determined by its specific heat capacity and density. So, a very dense material will exhibit comparatively small changes in temperature for any given amount of heat energy input. Finally, the rate at which heat is conducted to the bottom of the rock will be driven by its thermal conductance (a material characteristic) as well as the rock thickness. These latter reasons are the underlying mechanisms driving the different temperatures recorded by Huey et al under rocks of different thickness.

None of this invalidates the study, but instead it urges caution in statements that imply that one rock type is “hotter” than another (line 62), or that these differences can be untangled purely using experimental approaches.

Along these lines, the paragraph that begins on line 36 needs clarification. There are now quite a few studies that have measured baseline conditions of rock temperature on shorelines using temperature loggers, see for example a host of papers by Lima and Seabra. Realistically, however, one can’t measure temperatures on every shore or indeed on any microhabitat on any given shoreline. To try to tackle the latter, one needs to use heat budget models and again there is now a pretty long history of this from research groups such as mine (Helmuth), Wethey, Lima and Denny. The quite valid criticism of these modeling approaches- and where studies such as the one presented here provide significant value- is that they have almost universally made assumptions about rock type by, for example, assuming that an entire shore is made of granite or concrete. This study shows that rock type (for likely all of the reasons above) can potentially play an important but neglected role in driving patterns of heterogeneity in temperature.

A few more specific points and questions:
Line 164. I’m not sure what the point of this was. Am I reading this correctly that rocks were only submerged for 20 minutes? This certainly would have wetted their surfaces, but since you allowed them to dry I assume you were trying to cool them to the same starting temperature, as would normally occur in an intertidal zone as the tide rolle out? From your data, can you confirm that they were all at the same starting temperature as one another at 9:00 at the start of each trial? If so (or if not), this should be mentioned.

Lines 300-307. How did you quantitatively distinguish between a gradient and a mosaic? The examples shown in the figure seem pretty straightforward, but I am guessing there were some intermediates. Do you do some sort of spatial regression or other method of assessing variance from one spot to the next?

Line 327 Be a bit careful in the interpretation of quartzite- it is likely that at least some of the differences you observed were due to differences in emissivity, which would have falsely shown up as heterogeneity in the IR camera. If you still have the rocks, an interesting experiment to ground truth this would be to record patterns with the camera, and then again with something like a thermocouple to verify that you were recording actual differences in temperature and not just differences in surface emissivity (which would have shown up as apparent temperature differences in the IR camera which may or may not be real).

Line 476 “Air temperature also affected boulder temperature behaviour, with the hottest boulder temperatures measured on the hottest sunny days.” This actually is not evidence of the role of air temperature. Air temperature is driven indirectly by surface temperature, which in turn is driven by solar radiation. So, the reason why air temperature is hot on sunny days is because surface temperature is hot, not the other way around.

Line 495. I’m not sure what you mean here: “While within-microhabitat temperature differences have been described previously in terrestrial habitats (e.g. Huey et al. 1989), these results are novel for intertidal rocky substrata.” There have been a bunch of studies measuring small-scale (cm) differences on intertidal rock surfaces, driven largely by (or assumed to be driven by) differences in shading, which create the microclimates. Here you have shown that these microclimates can be influenced by rock type, but the patterns observed on top of the rock still are a combined influence of small-scale micro shading and rock type- this study doesn’t decouple those (e.g. even sandstone has small-scale features which can drive microshading). See for example Choi et al. 2019 Conservation Physiology and Pincebourde and Woods 2020 Current Opinion in Insect Science. Bottom line, I don’t think you are using “microhabitat” consistently. A single boulder is not a “microhabitat” but rather contains lots of microhabitats.

Lines 510-511. Nope, there actually have been quite a few studies looking at differences in physiology among intertidal organisms exposed to different microclimates.

Lines 524-532 Again here you are ignoring the role of small-scale structures/surface complexity in driving mosaics. Just because you chose rocks with limited surface texturing doesn’t mean these don’t exist, so it is a big stretch to state “Given predictions of an increased frequency of heatwaves and generally hotter air temperatures associated with global climate change (IPCC 2013), hot boulder surfaces with limited heterogeneity may be observed more often.” I agree, figuring this out is important, but you really can’t make this extrapolation based on this study.

Line 541-542: “cooler rock types that only warm small amounts when emersed may offer thermal refuges to thermally-sensitive species.” As above, be more cautious in saying that one rock type is “cooler” than another. For example, rocks with a high density and high specific heat capacity will that more slowly, but they will also cool more slowly and during long exposures (especially high on the shore) they may ultimately heat to higher temperatures.

Line 586: “However, our results, in conjunction with those published previously for the same siltstone boulders (Chapperon and Seuront 2011a; Chapperon et al. 2013), suggest that the magnitude of temperature relief that biota experience under boulders is actually quite small (<5 °C) when compared to temperature differences of up to 25.5 °C between sun-exposed and sun-protected microhabitats elsewhere on rocky seashores (e.g. Denny et al. 2011; Lathlean et al. 2012).” This is extrapolating well beyond your data. True, you only observed a 5C difference but then you also only recorded temperatures after 4 hours and many exposures can be much greater in the high intertidal zone. And, as above, rock thickness probably plays a much bigger role than rock type.

Lines 630-642. I would delete this paragraph. All it basically says is that the sun heats rocks and that is hardly surprising.

Lines 653-659. Again, I’d suggest cutting. This study certainly informs such studies, but these data alone from this one site likely won’t provide the baseline data needed to design experiments on different shores.

---

## Round 0.2 · Minor Revisions

I am pleased to find both referees are also pleased with the revised version. One makes some good suggestions for improvements that will improve the clarity and messages from the paper. Please consider these and submit a final version.

Reviewer 2 ·

Basic reporting

n/a

Experimental design

n/a

Validity of the findings

n/a

Additional comments

I am happy with changes and clarifications made to revised manuscript

·

Basic reporting

The paper is well written and while still a bit long is much better than the first revision. Nice job in responding to reviewer comments.

Experimental design

There are still some issues with the generic value of emissivity used, and of the use of the term "gradient" which has a very real physical meaning, as opposed to the qualitative way it is used here. I have made some suggestions in the draft.

Validity of the findings

Fine

Additional comments

See attached

---

## Round 0.3 · accepted · Accept

Thank you for revising the text to clarify it based on the referee's suggestions and omitting terms that could be confusing or misleading.